# Translating Biomarkers of Cholangiocarcinoma for Theranosis: A Systematic Review

**DOI:** 10.3390/cancers12102817

**Published:** 2020-09-30

**Authors:** Imeshi Wijetunga, Laura E. McVeigh, Antonia Charalambous, Agne Antanaviciute, Ian M. Carr, Amit Nair, K. Raj Prasad, Nicola Ingram, P. Louise Coletta

**Affiliations:** 1Leeds Institute of Medical Research, Wellcome Trust Brenner Building, St James’s University Hospital, Leeds LS9 7TF, UK; i.wijetunga@nhs.net (I.W.); 20mcveigh@gmail.com (L.E.M.); antonia.charalambous11@gmail.com (A.C.); agne.antanaviciute@ndm.ox.ac.uk (A.A.); i.m.carr@leeds.ac.uk (I.M.C.); nair.amit@doctors.org.uk (A.N.); n.ingram@leeds.ac.uk (N.I.); 2Department of Hepatobiliary and Transplant Surgery, St. James’s University Hospital, Leeds LS9 7TF, UK; raj.prasad@nhs.net

**Keywords:** biomarkers, cholangiocarcinoma, theranosis, therapy and diagnosis, biomarker selection

## Abstract

**Simple Summary:**

Bile duct cancers are rare cancers that have poor prospects and limited treatment options. Recently, significant advances have been made in the field of nanomedicine which has allowed new approaches to the diagnosis and treatment (i.e., theranosis) of human diseases. To develop nanomedicines that could earmark or target bile duct cancer, specific proteins (or biomarkers) that are present in bile duct cancer but absent in normal tissues are required. We conducted a systematic search of the published literature for bile duct cancer biomarkers that would be suitable for theranosis. Specialist bioinformatics tools were used to help categorize the resulting data set. To select the most promising biomarkers from the search, biomarkers were ranked according to a theranosis-scoring-system and then evaluated in detail. The biomarkers identified using this approach have the potential to promote targeted nanomedicine-based systems to treat bile duct cancers.

**Abstract:**

Cholangiocarcinoma (CCA) is a rare disease with poor outcomes and limited research efforts into novel treatment options. A systematic review of CCA biomarkers was undertaken to identify promising biomarkers that may be used for theranosis (therapy and diagnosis). MEDLINE/EMBASE databases (1996–2019) were systematically searched using two strategies to identify biomarker studies of CCA. The PANTHER Go-Slim classification system and STRING network version 11.0 were used to interrogate the identified biomarkers. The TArget Selection Criteria for Theranosis (TASC-T) score was used to rank identified proteins as potential targetable biomarkers for theranosis. The following proteins scored the highest, CA9, CLDN18, TNC, MMP9, and EGFR, and they were evaluated in detail. None of these biomarkers had high sensitivity or specificity for CCA but have potential for theranosis. This review is unique in that it describes the process of selecting suitable markers for theranosis, which is also applicable to other diseases. This has highlighted existing validated markers of CCA that can be used for active tumor targeting for the future development of targeted theranostic delivery systems. It also emphasizes the relevance of bioinformatics in aiding the search for validated biomarkers that could be repurposed for theranosis.

## 1. Introduction

Cholangiocarcinomas (CCA) are a group of cancers of the biliary system which are usually diagnosed late, often with a dismal prognosis [1,2,3,4]. The most commonly used classification of CCA is anatomical and describes peripheral or intra-hepatic (iCCA), hilar or perihilar (pCCA), and distal (dCCA); the latter two being subtypes of extra-hepatic CCA (eCCA) [2]. There is geographical variation in the incidence of CCA with higher incidence in Eastern countries compared to Western nations where this is a rare, sporadic cancer [5,6]. This cancer is especially prevalent in Thailand, where CCA is a significant national health burden due to the prevalence of the liver fluke, which is an etiological factor [7]. The only curative option for all subtypes involves aggressive surgical resection with or without adjuvant therapy; however, only a minority are eligible [2,8]. In unresectable disease, the median overall survival even with palliative chemotherapy remains less than twelve months [9]. To improve these poor outcomes, discovery of new diagnostic markers, development of targeted therapies to improve the translation of new strategies such as nanomedicine-based technology, and predictive markers to determine response to therapy should be explored. There has been a recent push towards the identification of genetic drivers of CCA progression and this may reveal specific markers which could expedite the development of more effective and individualized therapies [10,11].

Theranostics is a new field of medicine that combines targeted therapies with diagnostics [12,13,14]. The term has been coined to define the ever evolving ‘precision medicine’ approach whereby diagnosis, drug delivery and monitoring treatment response can be combined and tailored for each patient. 

Over the last decade, significant advances have been made in the field of nanomedicine and its potential applications for the diagnosis and treatment of human diseases [15,16,17]. Concomitant use of nanoscale particles for theranosis has been evaluated in malignant and non-malignant conditions [18,19,20,21]. Active tumor targeting to upregulated cancer cell receptors or tumor specific markers by nanoparticles that have been surface-functionalized can potentially increase the specificity of nanoparticle-based therapy. Active targeting relies on a “molecular ligand” that is upregulated in tumor tissue compared to normal tissues to increase tumor specificity of nanoparticle uptake and reduce off-site delivery/toxicity [22]. 

A recent systematic review on the application of active targeting nanoparticle delivery systems in cancer therapy did not show any pre-clinical studies on CCA [23]. This reflects the general scarcity of research efforts into novel therapeutics for this rare cancer. 

Notable advances have been made in the discovery of biomarkers associated with CCA over the last decade. In this study, we aimed to systematically interrogate the literature for published studies in CCA for the purpose of identifying suitable biomarkers for active tumor targeting which in future may aid in the development of a theranostic delivery platform combining molecular imaging and therapeutic drug delivery. The methods used here may act as a blueprint to identify suitable biomarkers for theranosis that are applicable to other cancers or disease states. 

## 2. Materials and Methods

### 2.1. Systematic Search Strategy for Theranostic Biomarkers in CCA

The present review was conducted according to the Preferred Reporting Items for Systematic reviews and Meta-Analyses (PRISMA) guidelines and recommendations. Electronic searches were performed using MEDLINE/EMBASE from January 1996 to September 2019. Two strategies were used to systematically search and identify immunohistochemical and proteomic biomarker studies of CCA using the following search terms; Cholangio* OR Bile duct* OR Biliary tract OR Klatskin AND Cancer OR Adenocarcinoma AND Immunohistoch* as well as Cholangio* OR Bile duct* OR Biliary tract OR Klatskin AND Cancer OR Adenocarcinoma AND Proteome OR Proteomics. The search strategy included all types of CCA biomarker studies (i.e., diagnostic, prognostic, predictive biomarkers, as well as those comparing biomarkers between other liver tumors and disease states). The search identified 4560 studies. The PRISMA flow diagram shown in Figure 1 depicts the search strategy and the two main search phases of this systematic review. Three authors (Imeshi Wijetunga (I.W), Laura E. McVeigh (L.E.M) and Antonia Charalambous (A.C)) were involved in phases 1 and 2 of the screening and data extraction according to the agreed protocol. Any queries were resolved by discussion among I.W., L.E.M., and A.C.

### 2.2. Selection Criteria

Stage 1 of the screening process covered the titles and abstracts of the identified studies. This revealed that 3859 did not meet the inclusion/exclusion criteria. The inclusion criteria used to screen all identified articles were (1) all markers discovered in bile or tissue from CCA patients by proteomic analysis and subsequently validated by immunohistochemistry (IHC) in at least 30 CCA specimens or IHC studies that reported upregulation of biomarker (greater than 20% positive tumor tissues and/or greater than 20% upregulation in tumor tissues), and (2) for IHC validation, the inclusion of at least 30 iCCA/eCCA/pCCA specimens was required (similar to the cut off used by a recent systematic review by Wiggers et al. [24]). The exclusion criteria applied to all identified studies were (1) studies only involving cancer cell lines with no tissue validation; (2) circulating markers in blood and urine were excluded unless they were also upregulated in the cancer tissue of origin by proteomic or IHC methods; (3) upregulation in normal tissues (greater than 20% positive normal tissues in a cohort and/or greater than 20% of normal tissues upregulation) as determined using Human Protein Atlas [25]; (4) downregulated biomarkers in cancer tissue; and (5) review papers, letters, editorials, conference abstracts, and case reports. All publications were limited to those involving human patients and in the English language. 

### 2.3. Data Extraction

For all included biomarkers, data were extracted on the study population demographics, tumor site, reported percentage upregulation or percentage positive presence, subtypes of CCA included, cellular location of tumor marker evaluated, whether tissue microarrays (TMAs) or whole sections were used, if a control group of tissues was included, percentage expression of biomarker in normal tissues, number of investigators assessing tissue, and if they were independent and/or blinded. 

### 2.4. Bioinformatics

The genes coding for the selected proteins were subjected to gene ontology analysis using PANTHER GO-Slim classification system [26] and categorized based on the tissue compartment of expression using the freely available online resource [27]. Biomarkers that were validated in two or more studies were included in the final analysis and interrogated in more detail by targeted literature searches. The STRING network (version 11.0) [28] was also used to further interrogate the final shortlist of biomarkers for theranosis to ascertain any known associations between the more promising biomarkers [29].

## 3. Results

### 3.1. Literature Search

A total of 701 studies, which included 1190 biomarkers, were selected for full text review in stage 2 of the screening process (see Figure 1 for PRISMA flow diagram). This involved evaluating each reported biomarker against the inclusion/exclusion criteria as well as scrutiny of the Human Protein Atlas annotated protein level data for normal tissue presence of each biomarker [25]. One-thousand-and-twenty-four biomarkers were excluded for not meeting inclusion/exclusion criteria, and the main reason for exclusion was the upregulation of the biomarker in normal tissues. In cases where no annotated protein level data was reported in the Human Protein Atlas, biomarkers were included in the data extraction unless the study reported upregulation in normal tissues which fulfilled the exclusion criteria (i.e., greater than 20% positive normal tissues in cohort and/or greater than 20% of normal tissues with upregulation).

One-hundred-and-sixty-six candidate biomarkers that met the inclusion/exclusion criteria were identified and included in the data extraction table summarized in Appendix A. Once replicate biomarkers were excluded, 83 different biomarkers were identified, of which only 47 had annotated protein level data reported in the Human Protein Atlas. Of these 47 biomarkers, only 16 were evaluated in two or more biomarker validation studies and thus selected for further interrogation. These 16 included angiopoetin 1 (ANGPT1), angiopoetin 2 (ANGPT2), carbonic anhydrase 9 (CA9), cadherin 17 (CDH17), caudal type homeobox 2 (CDX2), Claudin 18 (CLDN18), epidermal growth factor receptor (EGFR), matrix metalloproteinase 7 (MMP7), matrix metalloproteinase 9 (MMP9), alpha-1 antitrypsin (SERPINA1), secreted frizzled related protein 1 (SFRP1), solute carrier family 2 member 1 (SLC2A1), trefoil factor 1 (TFF1), transforming growth factor beta-1 (TGFB1), tenascin C (TNC), and p53 (TP53).

### 3.2. Gene Ontology of Selected Candidate Biomarkers

Figure 2 shows the PANTHER GO-Slim [30] classification of the 16 genes corresponding to the identified proteins into the three gene ontology domains: molecular function, cellular component, and biological process. Figure 2a categorizes the molecular function of the product of these genes. Twelve out of sixteen are involved in binding such as protein binding, chromatin binding, heterocyclic compound binding or ion binding. Four out of sixteen have catalytic activity either hydrolase (MMP9, MMP7, and SERPINA1) or transferase (EGFR). Others such as SERPINA1 and TGFB1 are molecular function regulators. In terms of cellular component, the majority of the gene products in the gene set were present in the extracellular region (Figure 2b). EGFR, CDH17, SFRP1, and CLDN18 are present in the plasma membrane. 

TP53 and CDX2 are transcription factors present in the nucleus. Twelve out of sixteen are involved in cellular processes (Figure 2c) such as SERPINA1 and TGFB1 in cellular macromolecule metabolic processes; MMP9 and MMP7 in cellular component organization; and ANGPT1, ANGPT2, EGFR, and TP53 in signal transduction. 

The PANTHER Version 14 Go-Slim categorization [30], which is a curated form of the entire gene ontology database, did not include *CA9*, *SLC2A1*, and *TNC*. These were interrogated from the PANTHER complete gene ontology database. In summary, CA9 has carbonate dehydratase activity, is involved in bicarbonate transport, and has membrane localization. SLC2A1 is a transmembrane glucose transporter, which is involved in response to hypoxia and also has a membrane localization. TNC is an extracellular matrix structural protein that negatively regulates cell adhesion and is involved in extracellular matrix organization.

### 3.3. Selection of Biomarkers for Theranosis

A scoring system for the selection of potentially targetable biomarkers for imaging in colorectal cancer (TArget Selection Criteria or TASC score) has been reported by van Oosten et al. [32]. However, there are no published scoring tools for the selection of biomarkers incorporating a therapeutic aspect. Despite overlap with imaging biomarker characteristics, theranostic biomarkers additionally require very low or absent levels in normal tissue. A theranostic biomarker scoring system, which is a modified version of the TASC score [32] named TASC-theranosis score (TASC-T), was therefore developed. 

The sixteen selected candidate biomarkers were evaluated using TASC-T scoring criteria shown in Table 1. Briefly, points of different weightage were awarded to each biomarker for the following criteria; extracellular location, >20% positivity in tumor tissue, tumor to normal tissue level ratio >10, percentage positivity or upregulation reported, previous application as an imaging or theranostic biomarker either in pre-clinical or clinical studies, and if a ligand for the biomarker is available for clinical use. All biomarkers included in this review scored 4 points automatically as >20% upregulation was a selection criterion. The maximum score possible was 22 and biomarkers were considered to have theranostic potential if they scored 17 or higher (similar to TASC score described by van Oosten et al. [32]). Table 2 shows the TASC-T scoring for the 16 selected candidate biomarkers. A score of 17 or higher was awarded to MMP9, CLDN18, TNC, CA9, and EGFR. 

**Table 1 cancers-12-02817-t001:** Potential targetable biomarkers for theranosis (TASC-T) scoring criteria.

No.	Parameter	Score
1	Extracellular or membrane localisation of biomarker	5
2	>20% positivity in tumor tissue	4
3	Tumor to normal tissue ratio >10	3
4	Percentage positivity or upregulation in tumor tissue	
	>90%	6
	70–90%	5
	50–69%	3
	<49%	0
5	Previous application to imaging	
	Preclinical	1
	Clinical	2
6	Ligand in human trials	1
Total		22

**Table 2 cancers-12-02817-t002:** TASC-T scoring of 16 selected biomarkers validated in ≥2 studies.

No.	Tumor Biomarker	Extracellular	Membrane	Highest % Positivity/Upregulation Reported in CCA	Ligand in Human Trials	Previous Use in Pre-Clinical or Clinical Imaging	TASC-T (Max 22)
1	ANGPT1	Yes	No	43.7	No	No	9
2	ANGPT2	Yes	No	57.6	No	No	9
3	CA9	No	Yes	85	Yes (Iodine-124 labeled cG250)	Yes [33]	17
4	CDH17	No	Yes	52.9	No	No	15
5	CDX2	No	No	60	No	No	7
6	CLDN18	No	Yes	90	Yes (Claudiximab, Zolbetuximab)	No	19
7	EGFR	Yes	Yes	75	Yes (Cetuximab, Panitumumab)	Yes [34]	17
8	MMP7	Yes	No	80	No	Yes [35]	15
9	MMP9	Yes	No	67	Yes (Andecaliximab)	Yes [36]	19
10	SERPINA1	Yes	No	57	No	No	12
11	SFRP1	Yes	Yes	60	No	No	12
12	SLC2A1	No	Yes	52	No	No	12
13	TFF1	Yes	No	98.4	No	No	15
14	TGFB1	Yes	No	47	Yes (Fresolimumab)	Yes [37]	15
15	TNC	Yes	No	63.9	Yes (Neuradiab, Tenatumomab)	Yes [38]	18
16	TP53	No	No	84	No	Yes [39]	13

The STRING database version 11.0 [28] was used to ascertain any protein–protein interactions between the top five selected proteins as shown in Figure 3. This revealed that there are associations between MMP9, CA9, and EGFR as well as TNC and EGFR in terms of being reported together in published studies. There was also evidence of co-presence of TNC and EGFR as well as CA9 and EGFR. An experimentally determined association between TNC and EGFR has also been established. CLDN18, on the other hand, did not have any known associations with the other four proteins being investigated.

### 3.4. Selected Biomarkers for Theranosis in CCA

The biomarkers scoring 17 or higher in the TASC-T score, namely, MMP9, CLDN18, TNC, CA9, and EGFR, were selected for in-depth analysis of their potential as theranostic biomarkers in CCA. The study characteristics of the selected published studies of protein presence of these five biomarkers in CCA are summarized in Table 3. Three out of these top five candidate biomarkers have only been evaluated in two studies. Even in combination, CA9, CLDN18, and TNC have only been evaluated in 548 Eastern tissue specimens, 76% (417/548) of which were in iCCA tissues. MMP9 and EGFR have been evaluated in multiple studies, but a wide range of positivity has been reported. This may be due to difference is scoring methods. Although these two biomarkers have been evaluated in 1793 tissues specimens in total, over 50% (936/1793) were evaluated in iCCA tissues only. 

#### 3.4.1. Matrix Metalloproteinase 9 Presence in CCA

Matrix metalloproteinase 9 (MMP9) has been evaluated in eight studies that met the inclusion/exclusion criteria for this review [40,41,42,43,44,45,46,47]. Three-hundred-and-two iCCA and 334 eCCA (including 120 pCCA tissue) have been evaluated with reported MMP9 positive ranges of 45.6 to 62.5% and 47.3 to 58%, respectively. Only two studies commented on non-neoplastic biliary epithelium which was reported as faint or absent [40,41]. All eight studies investigating MMP9 involve Eastern tissue specimens from Korea, China, Japan, and Thailand. MMP9 scored 19/22 on the TASC-T score.

#### 3.4.2. Claudin 18 Presence in CCA

Claudin 18 (CLDN18) was reported in two Japanese studies which included a total of 110 iCCA tissues and 131 eCCA tissues [48,49]. Keira et al. [48] reported CLDN18 upregulation on the basolateral surface of tissues with absence in the normal biliary epithelium. It was not possible to ascertain the percentage positivity or upregulation in the 59 tissues included in this study. Shinozaki et al. [49] reported 43% positive samples in 83 iCCA tissues and 90% positive samples in 99 eCCA tissues. The annotated protein level of CLDN18 in normal tissues according to the Human Protein Atlas is restricted to the stomach with an “enhanced” score for reliability, which is the highest awarded [25]. CLDN18 scored 19/22 on the TASC-T score.

#### 3.4.3. Tenascin C Presence in CCA

Similar to CLDN18, Tenascin C (TNC) was also reported in two Japanese studies. However, all 109 tissue samples were from iCCA patients [50,51]. Neither of these studies described TNC in normal liver tissue. The annotated protein level data of TNC is negative in all 44 tissues investigated by the Human Protein Atlas [25] although the reliability of this pattern is “approved” which is one category higher than the lowest “uncertain” category. Interrogating the primary data for the two TNC antibodies used by the Human Protein Atlas, one (CAB004592, Santa Cruz Biotechnology) reported weak to moderate immunoreactivity of most tissues but pancreas, liver, central nervous system, and lymphoid tissues were negative. Immunoreactivity with the second antibody (HPA004823, Sigma-Aldrich) reported strong positivity in seminiferous ducts and weak to moderate positivity in glandular epithelia. TNC scored 18/22 on the TASC-T score.

#### 3.4.4. Carbonic Anhydrase 9 Presence in CCA

Two studies [52,53] have explored Carbonic Anhydrase 9 (CA9) in CCA tissues which included 198 iCCA tissue samples in total and reported 85% upregulation in one study and 44.7% in the other (Table 3). Neither of these studies commented on CA9 presence in adjacent normal liver tissue. No studies that met the selection criteria for this review have evaluated CA9 in eCCA tissues. The annotated protein level data of CA9 in normal tissues display moderate to upregulation in stomach, duodenum, small intestine, and gall bladder [25]. CA9 scored 17/22 on the TASC-T score.

#### 3.4.5. Epidermal Growth Factor Receptor Presence in CCA

This review included 15 different studies which report Epidermal Growth Factor Receptor (EGFR) presence in CCA [50,54,55,56,57,58,59,60,61,62,63,64,65,66,67]. Ten out of fifteen studies investigating EGFR involve Eastern tissue specimens from Japan, China, Korea, and Thailand with 5/15 originating from USA, Brazil, Italy, and Germany.

In 402 tissue specimens of iCCA, the reported positivity of EGFR ranged from 26 to 100%. Of the 634 eCCA specimens, 18–79% were reported as positive (Table 3). In 121 pCCA tissues and 173 CCA of unspecified subtype, 28.6–55% of tissues were reported as positive. The proportion of CCA tissues positive for EGFR is higher in iCCA compared to eCCA in the studies included in this review. This was also the case in the four studies that included both iCCA and eCCA tissues and reported EGFR results separately [56,57,58,61].

Of these 15 studies, only one study [57] commented on normal liver and biliary tract tissue presence of EGFR as positive for EGFR immunoreactivity in all normal cholangiocyte and hepatocyte membranes. Low levels of EGFR have been demonstrated in normal tissues such as liver, skeletal muscle, and skin [25]. EGFR scored 17/22 on TASC-T.

## 4. Discussion

This review has highlighted that there have been a vast number of biomarker discovery and validation studies in CCA. Herein, we describe a method of systematic review of literature for the identification and selection of biomarkers for theranosis that is applicable to other cancers (Figure 4). Although five biomarkers highlighted in this review have potential for being theranostic targets, most of them lack robust tissue validation data in CCA. CA9, CLDN18, and TNC have only been validated in two separate studies that met the inclusion criteria for this study, whereas MMP9 and EGFR have been investigated by multiple research groups. We will now discuss the potential of each of the five protein targets that could be used as a CCA biomarker for theranosis, describing previous relevant studies and remarking on their suitability.

### 4.1. Matrix Metalloproteinase 9

MMP9 upregulation and its negative association with prognosis in cancers such as gastric cancer has been reported and a monoclonal antibody aimed at MMP9 inhibition is currently in human trials [68]. A clinical trial commenced in 2016 comparing andecaliximab (also known as GS-5745), which is an MMP9 inhibitor, as monotherapy and in combination with anti-cancer agents in Japanese participants with gastric or gastroesophageal junction adenocarcinoma is currently awaiting results (ClinicalTrials.gov, NCT02862535). This is one of five completed trials that evaluated andecaliximab in solid tumors, but none have yet published results. None of these trials included patients with CCA. Although a humanized monoclonal antibody is available, to our knowledge there has not been any human studies published using MMP9 as an imaging or theranostic biomarker. The structure of MMP 9 is shown in Figure 5a.

Recently, Hakimzadeh et al. [36] developed a MMP9 and MMP2 targeted radiolabeled ligand for the purpose of imaging atherosclerotic lesions. On ex vivo measurement of radioactivity, they demonstrated increased signal in mouse aortic tissue with atherosclerotic lesions and subsequently confirmed MMP9 and MMP2 immunoreactivity on these tissue sections. They described the radiolabeling procedure as challenging with an insufficient yield of radiolabeled product [36]. This may be the reason the authors do not report any in vivo imaging. Although this ligand may be useful for imaging MMP9 expressing cancers, the lack of MMP9 specificity of the ligand could result in low signal-to-noise ratio, as MMP2 presence in normal tissues is more widespread compared to MMP9 [25].

None of the studies that met the inclusion/exclusion criteria for this review included patients from Western cohorts. A previous study at our institution investigated MMP9 in 54 pCCA specimens from a Western cohort of patients but this too did not meet the inclusion criteria for this study due to all adjacent non-cancerous liver control tissues being MMP9 positive [69]. This is in contrast to the reported low or negative normal liver tissue level reported by the Human Protein Atlas [25] and the 2/8 studies that reported low or absent MMP9 in normal biliary epithelium [40,41]. Whether this is due to differences in tissue handling and immunohistochemical processing or whether this is non-specific background labeling from the antibody utilized remains to be determined. If normal liver is indeed truly positive, MMP9 is unlikely to be a good theranostic target despite being highlighted as a potential candidate in this review.

### 4.2. Claudin 18

*CLDN18* has two main alternate splice variants [70] of which the alternate splice variant 2 expression is mainly seen in gastric epithelium whereas variant 1 is observed in lung tissue [71]. Sahin et al. [72] reported that protein coded by *CLDN18* splice variant 2, termed CLDN18.2, is present in several cancers including gastric (77% positive), pancreatic (80% positive), and esophageal (78% positive) cancers.

CLDN18.2 has been identified as a target suitable for therapeutic antibody development [72], and CLDN18.2-inhibiting monoclonal antibodies (e.g., claudiximab and zolbetuximab) are now in clinical trials for esophageal, gastric, and pancreatic adenocarcinoma. Two phase 3 trials using the addition of zolbetuximab or placebo to standard chemotherapeutic regimens are currently recruiting CLDN18.2 positive patients (ClinicalTrials.org: NCT03504397 and NCT03653507). However, a recent large immunohistochemical study of gastric cancer in a cohort of Caucasian gastric cancer patients only showed 42.2% (203/481) positive for the presence of CLDN18.2 [73].

Although published literature on CLDN18 in CCA is limited, one included study reported a difference in the presence in iCCA compared to eCCA. Shinozaki et al. [49] demonstrated a 90% positive presence in eCCA. Interestingly, other biomarkers present in gastric epithelium such as mucin 5AC (MUC5AC), mucin 6 (MUC6), and cytokeratin 20 (KRT20) have also shown to be higher in eCCA compared to iCCA [24].

The structure of CLDN18.2 as shown in Figure 5b has two predicted extracellular loops, the first of which includes the binding site for claudiximab. Tight junction proteins exist in the lateral apical surface between two epithelial cells. Whether the extracellular sites of tight junction proteins will be accessible to systemically delivered theranostic agents is not known. In favor of CLDN18.2 as a theranostic agent, in terms of its cellular location, it has been shown to be present throughout the cell membrane in normal physiology [71] and not just restricted to tight junctions. This may also be the case in cancerous cells as they are generally less tightly adherent to neighboring cells and have less well organized cellular barriers compared to normal tissue.

### 4.3. Tenascin C

TNC is a large protein that exists as a hexamer in the extracellular matrix [74]. Although abundant in embryonic tissues, its presence in adult tissues is limited to areas of inflammation, wound healing, and tumor growth [75,76,77]. This restricted pattern and upregulation during tumor growth has made TNC an attractive biomarker for developing targeted theranostics. Figure 5c shows a schematic of TNC structure as a monomer and hexamer.

This review has highlighted two studies which report TNC in iCCA. TNC in eCCA has not been reported. However, TNC presence in gastrointestinal malignancies such as pancreatic ductal [78,79], oesophageal [80], gastric [81], and colorectal [82] adenocarcinoma have been described.

With regard to TNC targeted imaging, Hicke et al. [83] demonstrated tumor targeting in a glioblastoma xenograft model using a TNC-specific RNA aptamer. They were able to image U251 glioblastoma xenograft tissue in vivo with Technetium-99m-labeled TNC aptamers using radionuclide imaging [83]. He et al. [84] developed a nanoparticle formulation of camptothecin prodrug conjugated with a cell penetrating peptide and TNC-targeted single-stranded DNA aptamer and evaluated this in a xenograft model of pancreatic ductal adenocarcinoma. They were able to demonstrate in vivo imaging of these particles and demonstrated therapeutic response in a xenograft model [84]. These successful applications of TNC-targeted theranostics using aptamers is promising as antibody mimetics have several advantages over antibodies including their lower cost, scalability, and more ethical synthesis as they do not require an intact host immune system for their production [85,86].

### 4.4. Carbonic Anhydrase 9

CA9 belongs to a large group of carbonic anhydrase enzymes which act as catalysts for the reversible reaction of carbon dioxide with water to bicarbonate and hydrogen ions [87,88]. Although this family of enzymes are ubiquitously expressed in humans, CA9 is restricted mainly to the gastrointestinal tract further upregulation is seen in gastric cancer and several other solid tumors [88,89]. The role of CA9 in the regulation of pH in the tumor microenvironment and induction of CA9 expression in response to tumor tissue hypoxia has been demonstrated [90]. 

Given the restricted presence in normal tissues and upregulation in solid tumors, CA9 has gained considerable interest as a novel tumor target [91]. Despite pre-clinical data supporting the use of CA9 inhibition as a therapeutic option for cancers, there have not been any human studies demonstrating its benefit [92]. The results of a phase I clinical study of a CA9 inhibitor in solid cancers [93] is still to be reported. 

In addition to low normal tissue distribution and upregulation in cancer, CA9 as a potential theranostic target has several other key advantages. It is a membrane-bound protein with an extracellular catalytic component as shown inFigure 5d. This makes it easily “visible” to systemically delivered theranostic agents without uptake into the cell. In addition, it has previously been used as an imaging biomarker [33]. Binding to CA9 and internalization of positron emission tomography (PET) tracers labeled with CA9 antibody have been demonstrated in a mouse xenograft model of renal cell carcinoma [94]. The other feature that could be exploited is its enzymatic activity, which is a desirable feature in an imaging [32] as well as theranostic biomarker. A theranostic agent that is intended to target CA9 could be designed to be activated by CA9 thus releasing its drug payload only when it has reached the tissue target.

Although restricted mainly to gastric mucosa, CA9 is present in normal tissue and has a role in pH regulation [95], and thus the potential side effects of CA9 as a theranostic target have to be considered. If this is to be explored in future in vivo studies, endpoint gastric mucosa histology should be assessed for potential off-target toxicity. Another disadvantage of CA9 as a theranostic target is that its presence in tumor tissues is induced by tissue hypoxia. If this is indeed the case, the delivery of a theranostic particle to the site of CA9 will be challenging, as systemically administered agents are likely to have low tissue penetration. 

### 4.5. Epidermal Growth Factor Receptor

Extensive research efforts have been focused on EGFR over the past two decades and anti-EGFR therapy in combination with conventional chemotherapy is now in routine practice for cancers such as metastatic colorectal cancer (mCRC) [96]. Cetuximab, which is a human/mouse chimeric monoclonal antibody, and the fully human monoclonal antibody panitumumab that inhibits EGFR were approved by the Food and Drug Authority (FDA) of the United States in 2004 and 2006, respectively [97]. Either cetuximab or panitumumab in combination with first-line chemotherapy has been evaluated in advanced biliary tract cancer (BTC) but neither has shown any significant improvement in progression-free survival or overall survival [98]. This suggests that inhibiting the EGFR pathway alone may not be beneficial or that it may be beneficial only in a subset of BTC, a difference these clinical trials were not sufficiently powered to detect.

A schematic structure of EGFR both in its inactive form and its dimerized active form is shown in Figure 5e. Cetuximab and panitumumab work by binding to a site on the extracellular domain of EGFR that partly occludes the binding site for native epidermal growth factor (EGF) [99,100]. 

Noninvasive imaging of EGFR status has been investigated and several EGFR radiolabeled tracers have been developed for PET and single-photon emission computed tomography (SPECT) [97,101]. In patients with mCRC, Van Helden et al. [101] were able to image tumor uptake of cetuximab labeled radiotracer, but this was not directly correlated with EGFR presence and did not predict treatment benefit with cetuximab. One confounding factor in this study was the pre-treatment of patients with unlabeled cetuximab to occupy non-malignant binding sites as invariably EGFR binding sites on the tumor could also be saturated which would have had an impact on subsequent imaging. This highlights that an ideal theranostic biomarker should have low or negative presence in normal tissues. In this ideal scenario where the presence of the biomarker of interest is truly negative in normal tissues, pretreatment would be unnecessary.

This review has highlighted 15 studies reporting the presence of EGFR in CCA. Although the scoring methods differed between studies, there was a trend towards upregulation of EGFR in iCCA compared to eCCA. This was still the case when only the four studies that reported EGFR in both iCCA and eCCA were considered [56,57,58,61]. This may suggest that EGFR may not be a useful theranostic maker in eCCA.

### 4.6. Challenges and Limitations

This systematic review highlighted the challenges and limitations of this type of study. The heterogeneity of the CCA biomarker studies and the scarcity of biomarker validation in multiple centers hampers the conclusions that can be drawn and makes meta-analysis of data impossible. The studies encountered in this review were mainly aimed at biomarker discovery of diagnostic, predictive, or prognostic biomarkers in CCA or those that could be direct therapeutic targets which were not always suitable candidates as theranostic biomarkers.

The five biomarkers described in detail in this study have theranostic potential in CCA as they can have both in vivo diagnostic (e.g., molecular imaging, mapping tumor margins and/or sites of metastatic disease) and therapeutic (e.g., targeted drug delivery using nanotechnology) applications. However, they are not CCA-specific, not necessarily detectable in body fluids such as blood or urine, and would not fulfill the role of early diagnostic biomarkers in the traditional sense.

The studies included often discussed proteins using different aliases, and therefore gene names had to be used to ensure that the data were extracted for the correct protein. Furthermore, there was significant variability in reporting of the results and scoring methods for the same biomarker making a comparison across studies very difficult. Often supplementary data was required to extract the necessary data and infrequently, studies had to be excluded as the percentage positive presence or upregulation in tissues could not be determined from the data provided.

The majority of studies involving histology specimens were performed in Eastern countries such as Thailand and Japan where CCA has a higher incidence due to etiological differences [6]. Although this geographical difference reflects the relative disease burden of CCA worldwide, the scarcity of tissue from Western cohorts makes it difficult for conclusions to be drawn about the applicability of these biomarkers for CCA theranosis in general. Hughes et al. [102] explored the differences in cell phenotypes in a cohort of liver fluke associated CCA (65 cases) in comparison to sporadic CCA (47 cases). They reported that liver fluke-associated CCA had more intestinal metaplasia phenotypes and was more likely to overexpress p53 in comparison to sporadic CCA. Whether other biomarkers expressed in liver fluke associated CCA are comparable to those in sporadic CCA and CCA that arises on a background of PSC seen in Western countries remains to be explored.

An attempt was made to assess the risk of bias of the studies including consideration of the criteria of the Newcastle–Ottawa Scale [103]. However, due to significant heterogeneity the only measure of bias that was included in the data extraction was the assessment of tissue. Studies were regarded as low bias if two or more blinded investigators independently scored the tissue sections. A significant proportion of studies did not utilize appropriate control tissues in parallel with CCA tissue for biomarker analysis, which was a significant deficiency. Where normal tissue controls were included, the immunolabeling pattern in normal bile duct tissue and hepatocytes lacked detailed description.

Although the Human Protein Atlas [25] was an exceedingly useful resource, 43% (36/83) of the potential candidate biomarkers did not have annotated protein level data in normal tissues. Even for those proteins that did, the reliability score was variable. The reliability score ranged from “enhanced”, “supported”, and “approved” to “uncertain”, reflecting the immunohistochemical data available using one or more antibodies as well as the RNA sequencing data [25]. Despite this, no biomarker was excluded based on the reliability score.

While guidance exists on reporting outcomes from tumor biomarker prognostic studies for IHC-based studies, i.e., REMARK guidelines [104], no such guidelines exist for tumor biomarkers in general. Therefore, the studies included in this review display significant deficiencies in reporting. Such guidance for reporting of all biomarker studies would result in a more consistent data set and hence improve the likelihood that these studies will have a wider impact on translational research. We recommend reporting presence of the candidate biomarkers in adjacent non-tumor tissue in all studies. We also recommend reporting of the cell type and cellular compartment where expression is observed. Another important aspect of reporting is the subtype of CCA from which the tissues originate as immunohistochemical differences have been observed in iCCA and eCCA [24].

The quest for ideal biomarkers that are specific to CCA tissue, which could then be actively targeted by drug-loaded nanoparticles, is still in its infancy. Given the heterogeneity of biomarker expression and the various anatomical and histological subtypes of CCA, individualizing the treatment strategy to develop customizable theranostic agents may be the way forward.

Two out of the five biomarkers (CLDN18 and CA9) highlighted in this review have also been reported as potential theranostic targets in gallbladder cancer [105]. EGFR and MMP9 have long been areas of research focus in cancer theranostics. TNC has also gained popularity as a biomarker for cancers as well as other conditions such as cardiovascular disease [106]. 

As demonstrated by this review, these five biomarkers with potential for theranosis in CCA have cross-specialty therapeutic uses allowing a broader application than just cancer for any theranostic nano-agent.

## 5. Conclusions

Here, we describe a method for systematic review of literature for the identification of theranostic biomarkers and assessment of biomarker characteristics according to their theranostic potential which involves the utilization of freely available bioinformatics tools to gather validated information. This method of identification and validation of theranostic biomarkers is therefore more widely applicable to other cancers and disease states. It is important to note that a high proportion of biomarkers identified in this systematic review dealt with those investigated for their prognostic value, which reflects the current published literature and perhaps a publication bias towards such studies in cancer. An early diagnostic biomarker that is also a therapeutic delivery target would be an ideal theranostic biomarker, and thus future research should concentrate on biomarker discovery of such biomarkers.

The five biomarkers with theranostic potential in CCA that were identified in this review have cross-specialty therapeutic applications. Knowing the scale of resources required to develop, validate, and translate biomarkers to clinical practice, collaborative efforts from different research groups will be essential in the future. The need for continued efforts to develop novel therapeutic strategies cannot be overemphasized for this rare group of cancers that have rising global incidence.

## Figures and Tables

**Figure 1 cancers-12-02817-f001:**
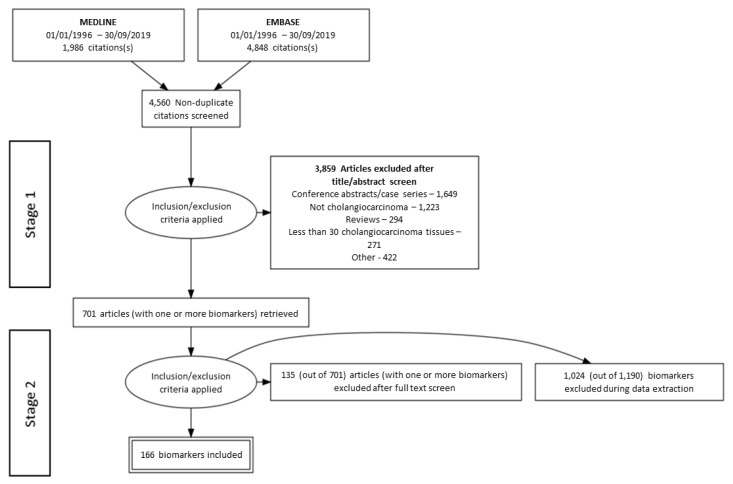
A flow diagram according to the Preferred Reporting Items for Systematic Reviews and Meta-Analyses (PRISMA) that maps the phases of the study selection process along with the number of records identified, excluded studies and/or biomarkers (and the reasons for exclusion), and biomarkers ultimately included in the systematic review.

**Figure 2 cancers-12-02817-f002:**
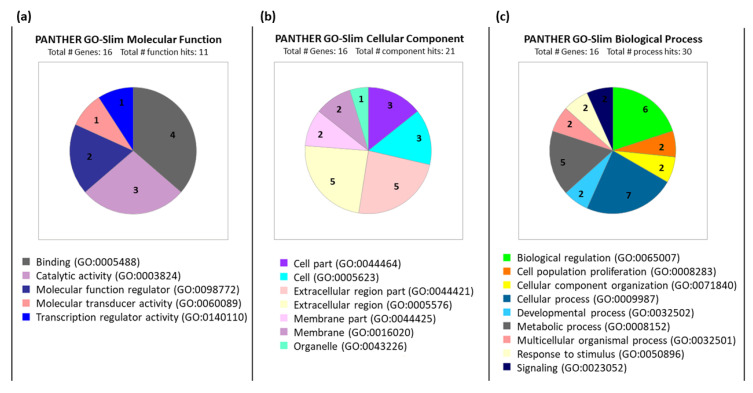
PANTHER classification of selected genes. PANTHER gene ontology classification of 16 selected genes based on (**a**) molecular function, (**b**) cellular component of expression, and (**c**) involvement in biological processes. Large-scale gene function analysis protocol for PANTHER Version 14 classification system used for this analysis [30,31].

**Figure 3 cancers-12-02817-f003:**
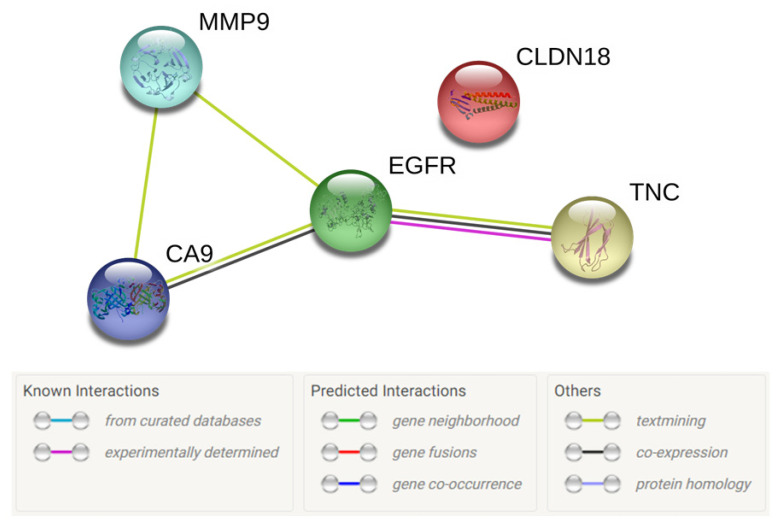
STRING network of the top five selected candidate biomarkers for theranosis in CCA. The green lines represent interaction between biomarkers as determined by text mining (automated, unsupervised search of PubMed for proteins mentioned together). This shows that there are associations between MMP9, CA9, and EGFR, as well as TNC and EGFR. The black line represents co-presence of biomarkers (CA9 and EGFR; TNC and EGFR). The pink line represents an experimentally determined association between TNC and EGFR. There are no known associations between CLDN18 and the other four selected candidate biomarkers.

**Figure 4 cancers-12-02817-f004:**
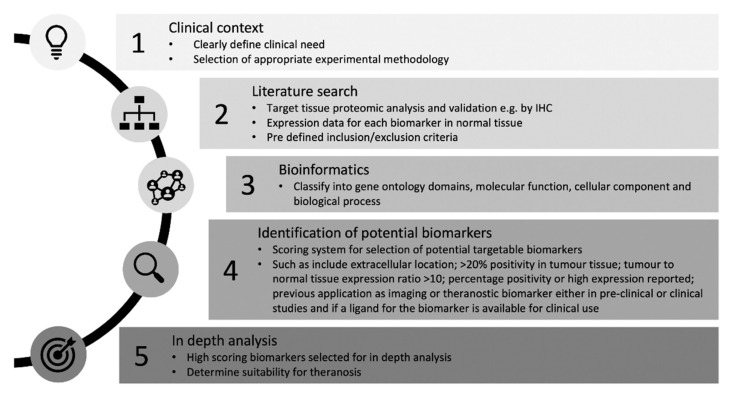
A blueprint to describe biomarker selection methodology for theranosis. A method of systematic review of the literature for the identification and selection of biomarkers for theranosis that is applicable to other cancers and disease states.

**Figure 5 cancers-12-02817-f005:**
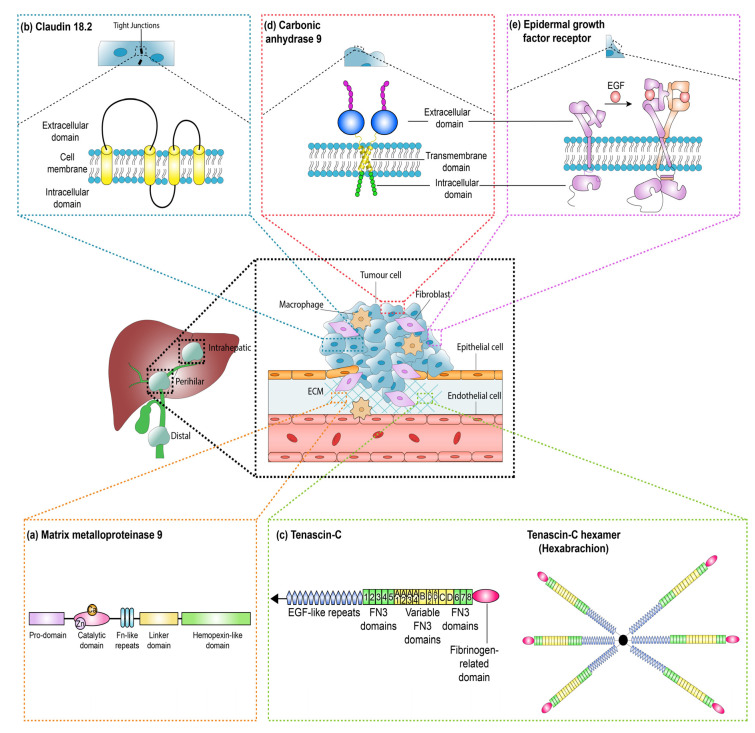
Schematic of the cellular location of MMP9, CLDN18.2, TNC, CA9, and EGFR. (**a**) MMP9 is present in the extracellular matrix (ECM). Schematic shows the different binding domains of MMP9. (**b**) Claudin 18.2 is present in the tight junctions and cell membrane. Schematic shows the two predicted extracellular loops and intracellular loop of Claudin 18.2. (**c**) Tenascin-C (TNC) is a large protein that exists as a hexamer in the ECM. Schematic (on the left) showing the five different domains of TNC from the N-terminus EGF-like repeats, fixed fibronectin domain, variable fibronectin domain, fixed fibronectin domain, and C-terminus fibronectin head. Schematic on the right shows the TNC hexamer, called hexabrachion. (**d**) CA9 is present in the cell membrane. (**e**) EGFR is present in the cell membrane. Schematic shows the inactive form of EGFR which dimerises on binding of Epidermal growth factor (EGF) to its active form.

**Table 3 cancers-12-02817-t003:** Study characteristics of the top five selected candidate biomarkers for theranosis in CCA.

Biomarker	Country	Site of Tumor	No.	% High/+ve Presence	Normal Tissue Control	Presence in Normal Tissues	Risk of Bias	References
MMP9 *	Multiple (mainly Eastern)	iCCA	302	45.6–62.5% +ve	Normal liver tissue	Weak or −ve	Low-High	[40,41,42,43,44,45,46,47]
		eCCA	214	47.3–58% +ve				
		pCCA	120	67.2–67.7% +ve				
CLDN18	Japan	iCCA	27	NR	Biliary epithelium	NR	High	[48]
		eCCA	32	NR				
	Japan	iCCA	83	43% +ve	Biliary epithelium	−ve	High	[49]
		eCCA	99	90% +ve				
TNC	Japan	iCCA	61	63.9% +ve	NR	NR	High	[50]
	Japan	iCCA	48	37.5% high	NR	NR	High	[51]
CA9	China	iCCA	113	85.0% high	NR	NR	Low	[52]
	Korea	iCCA	85	44.7% +ve	*n* = 4 normal liver	NR	High	[53]
EGFR *	Multiple (mainly Eastern)	iCCA	634	26.1–100% +ve	Normal liver tissue	+ve membrane	Low-High	[50,54,55,56,57,58,59,60,61,62,63,64,65,66,67]
		eCCA	402	18–79% +ve				
		pCCA	121	45.5% high				
		CCA	173	28.6–55% high				

* Data amalgamated from multiple studies and range of protein presence for each CCA subtype shown. NR–not reported. +ve–Positive.

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
