# Peer review of "Translating Biomarkers of Cholangiocarcinoma for Theranosis: A Systematic Review"

_cancers, 2020, doi:10.3390/cancers12102817_

Round 1
Reviewer 1 Report
In my opinion, this work is almost prepared for publication. The only question to be improved is the quality of the figures, that I still find they do not meet the requirements on resolution. In addition to this, a bigger size of font is desired in some cases that otherwise are quite difficult to read.
Author Response
Thank you for your comments.
The resolution problem is due to the size of the image on the edited manuscript and conversion to PDF. High resolution files of Figures 1, 4 and 5 have been uploaded to Zenodo (Available via the following link https://doi.org/10.5281/zenodo.4043373) if the editors wish to include high resolution images along with the online version of the manuscript, if published.
We hope this will resolve any issues with figure resolution and would satisfy the reviewer.
Reviewer 2 Report
Unhappy with reply to concern #1, which is unclear and not consistent with considerations made by authors in concern #2 on the characteristics of an ideal biomarker for a theranostic delivery platform. Authors must provide clear explanations on issue # 1 in the conclusions.
Author Response
Thank you for your comments.
We have provided further amendments to our original response to concern #1 and concern #2. Please see attached document for details.
We have also amended the conclusion to clearly address the concern as suggested (please see lines 506-511 in re-revised manuscript).

Reviewer 3 Report
Authors responded to every doubt arise after the previous sbmittion.
For this reason, I would like to suggest editors to accept in this form the manuscript revised.
Author Response
Thank you for your comments and acceptance of our revised manuscript. We appreciate your time and expert opinion.
Reviewer 4 Report
The authors did not address the comments:
"
- The authors indicate in inclusion criteria that biomarkers should be increased in CCA compared with “normal” tissue, however, according to the data presented in some cases the expression in control tissue is not reported and, when available, this normal tissue is not always the same; but sometimes is biliary epithelium, which seems the correct control, and in other cases is normal liver, where hepatocytes are the most abundant cells and is not indicated that the expression was compared with cholangiocytes. This could significantly affect the value of the selected biomarkers.
- There is an important heterogeneity in the information collected from different manuscripts and some include different CCAs, while others find differences in iCCA and extrahepatic CCA. Maybe the results should be presented separated? In addition, as indicated in table 3, the risk of bias is high for almost all the studies, which reduces the value of the selected biomarkers
- In line 145 the authors indicate that “the majority of the gene products in the gene set were expressed in the extracellular region”, that means that these genes are expressed in other cells different from cholangiocytes?
- Are the genes selected expressed in cholangiocytes, but not in hepatocytes?
- Some references regarding general characteristics of CCA are not very recent. Although the prognosis for CCA patients has not changed much, it would be better to include some recent references.
- The authors use a classification of CCA that is not the most accepted; it would be more correct to use iCCA, pCCA and distal CCA (dCCA) instead of IH-CCA, PH-CCA and EH-CCA.
Minor points:
- English needs to be revised
- In line 38, the authors indicate that “five-year survival even with palliative chemotherapy remains less than twelve months”. However, this is not correct, they should say that the median overall survival …. remains less than twelve months or that five-year survival …. is X%.
- In line 143 the authors indicate that EGFR has transferase activity. Is it possible to indicate a reference for that?
- In legend of figure 2, they should indicate what is included in a, b and c figures. Maybe is better to include different colors for biological regulation and cell proliferation in figure 2c.
- Page numbers are repeated.
- In section 4.2, there is a paragraph that starts with “Results from a clinical trial…” and finishes with “is currently awaiting results”
Author Response
Thank you for your comments.
These were exactly the same comments made on our original submission (cancers-857258) which we addressed point by point in our cover letter to the revised submission (cancers-931915 submitted 29.08.2020). Is it possible that Reviewer 4 has not seen the cover letter and the point by point responses that we submitted along with our revised manuscript (cancers-931915 submitted 29.08.2020)?
Please see the attached document that addresses all your comments. The line numbers mentioned in the reviewer's comments refer to the original submission (cancers-857258) but the author responses refer to the the re-revised version of the manuscript (cancers-931915 submitted 22.09.2020).

Round 2
Reviewer 2 Report
No further concerns on the revised manuscript.
Reviewer 4 Report
Acceptable and can be published
This manuscript is a resubmission of an earlier submission. The following is a list of the peer review reports and author responses from that submission.
Round 1
Reviewer 1 Report
Currently, we have not had a specific and strong biomarker(s) to detect the early insurgence of cholangiocarcinoma. Moreover, we miss for therapeutic and prognosis biomarker(s).
Authors would like to suggest five specific biomarkers to use as theranostic biomarkers. In my opinion, authors did some important mistakes.
- They have not evaluated expression in a similar tumour such as hepatocellular carcinoma, that is the first liver tumour for incidence.
- Authors have not performed analysis about differential expression of indicate biomarkers among different type of cholangiocarcinoma (intrahepatic, extrahepatic or perihilar). Many studies described the cholangiocarcinoma as a set of tumours differentiated by anatomical site of insurgence and cell of origin.
- Authors have not described expression levels of biomarkers in physiological condition and/or in disease that increase the risk of cholangiocarcinoma insurgence such as primary sclerosing cholangitis or HBV/HCV infection.
For these reasons, I consider the manuscript submitted not satisfactory for standards of journal.
Reviewer 2 Report
The review of Wijetunga and coll. aimed at identifying tissue biomarkers potentially applied to imaging studies to enable early diagnosis and treatment of cholangiocarcinoma (CCA), and thus candidate as potential diagnostic/therapeutic (theragnostic) biomarkers. Taking a dual approach, based on the PATNTHER Go-Slim classification system and STRING networking, the authors identified a first set of biomarkers that were then scored according to TASC-T to select five proteins of interest, CA9, CLDN18, TNC, MMP9 and EGFR. Although the topic is of strong interest, and the candidate biomarkers identified by the authors might be functionally relevant in CCA biology, the study is flawed by three bias at the conceptual level.
- Since most studies analyzed by the authors dealt with biomarkers investigated for their prognostic significance, it is difficult to figure out how they can be turned to some utility in the early detection of CCA.
- Again, most data were obtained from Asiatic studies where molecular profile of CCA caused by fluke infections seem to be quite specific and thus difficult to be applied to CCA of different etiologies. This is of utmost importance for pre-malignant conditions, like primary sclerosing cholangitis, which would most benefit of the approach proposed in this paper to develop surveillance programs. This is indeed a major gap in knowledge in CCA management.
- Although application of TASC-T criteria can be relevant for the diagnostic purposes of imaging studies, they do not meet true theragnostic needs. In fact, these criteria developed in the colorectal cancer setting, penalize biomarkers with nuclear expression (i.e. S100A4) that hold most promise for therapeutic targeting especially in CCA.
Reviewer 3 Report
In this work, the authors conduce a systematic review in order to identify putative biomarkers for theranostics in cholangiocarcinoma (CCA). After an exhaustive research and screening, 5 candidates are chosen: Matrix metalloproteinase 9 (MMP9), Claudin 18 (CLDN18), Tenascin C (TNC), Carbonic anhydrase 9 (CA9) and Epidermal growth factor receptor (EGFR).
The strengths of this investigation are, firstly, the interest per se of the topic, given that CCA is the less studied hepatic cancer; and secondly, the fact of taking advantage of already published information to get the best of it. This latest is especially important in my opinion, given that nowadays lot of information is generated by omics approached and metadata analysis and it is time to put such information together and confirm it validity and utility.
Comments:
1.- None of the 5 proposed biomarkers are CCA specific, so they only could be used as therapeutical targets and for monitoring treatments, just once the condition is diagnosed. Please, discuss this issue in the Discussion section.
2.- On the other hand, clinic currently focuses on non-invasive methods of diagnosis and monitoring. Apart from imaging approaches, liquid biopsy has special interest and value. So, I wonder if the 5 protein candidates can be detected in blood, for instance.
3.- In this line, I also wonder if these 5 proteins could be validated in blood of real patients by the authors.
4.- Figures have not good quality, please solve this.
5.- In the Figure 2, I would prefer pie chats with the number of genes (proteins) in each sector. It is much more visual and informative.
6.- The author use in excess the word “expression” to talk about proteins. Moreover, the term is not appropriated since only genes are expressed. Proteins are synthetized, regulated (up/down), detected, accumulate …etc. So, please revise this and adapt the text if necessary. Sometime just removing the term (expression) is enough to make sense.
7-. According to the typescript: please, try not to use repetitive words too close in the text (see comments on the pdf document).
8.- Be careful with the punctuation (see comments on the pdf document).
9.- Revise annotations in the pdf document.

Reviewer 4 Report
Wijetunga et al. performed a systematic review of specific markers for CCA with potential interest for theranosis. The idea is interesting, however, there are some aspects that need to be revised:
- The authors indicate in inclusion criteria that biomarkers should be increased in CCA compared with “normal” tissue, however, according to the data presented in some cases the expression in control tissue is not reported and, when available, this normal tissue is not always the same; but sometimes is biliary epithelium, which seems the correct control, and in other cases is normal liver, where hepatocytes are the most abundant cells and is not indicated that the expression was compared with cholangiocytes. This could significantly affect the value of the selected biomarkers.
- There is an important heterogeneity in the information collected from different manuscripts and some include different CCAs, while others find differences in iCCA and extrahepatic CCA. Maybe the results should be presented separated? In addition, as indicated in table 3, the risk of bias is high for almost all the studies, which reduces the value of the selected biomarkers
- In line 145 the authors indicate that “the majority of the gene products in the gene set were expressed in the extracellular region”, that means that these genes are expressed in other cells different from cholangiocytes?
- Are the genes selected expressed in cholangiocytes, but not in hepatocytes?
- Some references regarding general characteristics of CCA are not very recent. Although the prognosis for CCA patients has not changed much, it would be better to include some recent references.
- The authors use a classification of CCA that is not the most accepted; it would be more correct to use iCCA, pCCA and distal CCA (dCCA) instead of IH-CCA, PH-CCA and EH-CCA.
Minor points:
- English needs to be revised
- In line 38, the authors indicate that “five-year survival even with palliative chemotherapy remains less than twelve months”. However, this is not correct, they should say that the median overall survival …. remains less than twelve months or that five-year survival …. is X%.
- In line 143 the authors indicate that EGFR has transferase activity. Is it possible to indicate a reference for that?
- In legend of figure 2, they should indicate what is included in a, b and c figures. Maybe is better to include different colors for biological regulation and cell proliferation in figure 2c.
- Page numbers are repeated.
- In section 4.2, there is a paragraph that starts with “Results from a clinical trial…” and finishes with “is currently awaiting results”